# A Review: Development of Computer Vision-Based Lameness Detection for Dairy Cows and Discussion of the Practical Applications

**DOI:** 10.3390/s21030753

**Published:** 2021-01-22

**Authors:** Xi Kang, Xu Dong Zhang, Gang Liu

**Affiliations:** 1Key Lab of Modern Precision Agriculture System Integration Research, Ministry of Education of China, China Agricultural University, Beijing 100083, China; B20183080643@cau.edu.cn (X.K.); SY20183081402@cau.edu.cn (X.D.Z.); 2Key Lab of Agricultural Information Acquisition Technology, Ministry of Agricultural of China, China Agricultural University, Beijing 100083, China

**Keywords:** computer vision techniques, dairy cattle, image processing, lameness detection, visual locomotion scoring

## Abstract

The computer vision technique has been rapidly adopted in cow lameness detection research due to its noncontact characteristic and moderate price. This paper attempted to summarize the research progress of computer vision in the detection of lameness. Computer vision lameness detection systems are not popular on farms, and the accuracy and applicability still need to be improved. This paper discusses the problems and development prospects of this technique from three aspects: detection methods, verification methods and application implementation. The paper aims to provide the reader with a summary of the literature and the latest advances in the field of computer vision detection of lameness in dairy cows.

## 1. Introduction

Dairy cow lameness not only adversely affects dairy cow welfare and reduces dairy production but also degrades the reproductive capacity and increases the mortality rate [1,2,3]. According to the Research Report from Goldman Sachs, the average rate at which dairy cows go lame is 23.5%, which causes economic losses of 11 billion US dollars every year [4]. Lameness has a large impact on dairy farming and the national dairy industry development [5]. Researchers from various countries have investigated the incidence, economy and prevention of dairy cow lameness. The rate of lameness in Southern Brazil among the small-scale grazing dairy herds was more than 29.6%, and it was higher among Holstein and crossbred cows than Jersey cows [6]. The average lameness rate of cows was 18% in four European countries (France, Germany, Spain and Sweden) [7]; 36.8% among British dairy cows [8] and 34% among German and Austrian cows [9]. The lameness rates in dairy cows vary among countries due to different pasture production conditions and grazing methods [7], but cow lameness is common on farms everywhere.

This high lameness rate has a large impact on the dairy economy [10,11]. Lameness-related treatment costs rank second among common dairy diseases [12]. Each lame cow costs hundreds of dollars a year. The prices for treatment, testing, prevention, reduced milk production, reduced reproduction and increased mortality [11] are all considered. In fact, lameness is one of the most expensive diseases a dairy cow can contract [12]. It affects not only the economy but, also, the welfare of cows [13,14,15,16,17,18,19]. Therefore, many studies have focused on exploring the conditions that cause lameness and how to reduce it [20,21,22,23]. However, cow lameness is unavoidable under current farm conditions; thus, how to detect cow lameness in an accurate and timely manner is of great significance [24].

Lameness leads to behavioral changes in dairy cows when walking [25], as the animals reduce their speed, change their pace, arch their backs and bow their heads [26,27,28]. These behavioral changes are undertaken to compensate for pain and form important aspects of lameness detection [29,30,31,32]. The conventional method for lameness detection in dairy cows has been largely based on farmer observations [15,33,34]. To quantify the severity of lameness, some researchers have established a scoring system for cow lameness. The widely used five-point scoring system first proposed by Sprecher et al. [35] divides cow lameness into five stages based on changes in the cow’s back posture, standing posture and walking gait. Winckler and Willen [36] modified the scoring system of the five-point system by considering the step consistency, step size and load of a dairy cow’s gait. Later, Thomsen et al. [37] optimized the threshold judgment method for each grade of lameness to make the scoring system reliable. Trained professionals use the locomotion score while observing walking dairy cows, and this scoring system has become a common approach for assessing dairy cow lameness [38,39]. Nevertheless, the score is subjective, as it is influenced by the rater’s experience and perceptions [40,41], and it consumes manpower and material resources to obtain. On average, the number of lameness detections by farmers as identified by locomotion scoring is less 35% of the lame cows on a farm [42]. Additionally, its ability to detect mild lameness is poor [43]; thus, there is often a considerable delay between the onset of lameness and treatment [44,45,46]. Consequently, in recent years, electronic techniques have been increasingly introduced into the dairy industry that can detect the lameness of cows in a timelier manner and more accurately [47].

Automated lameness detection could provide useful cow- and herd-level information to address an information gap, particularly regarding mild and moderately lame cows [48]. The automatic methods of lameness detection broadly fall into three categories: kinematic, kinetic and indirect methods [28], and a major consideration in creating an automated lameness detection tool is choosing a type of sensor system upon which to focus [48]. The kinetic methods were mainly used to measure the ground reaction force or four-balance weight of each leg by a weighing platform or pressure-sensitive mat [49]. The indirect methods mainly used an accelerometer to detect the behavior and activity of cows [28]. The sensors used in these methods were contact sensors, which will be briefly introduced in this paper. A computer vision lameness detection system used the method of kinematics, which measures the geometry of movement without considering the forces that cause the movement. It had a moderate price and a noncontact information acquisition method [49]. The method demonstrated that the lame cows compared with the healthy cows will have a shorter stride length, longer stride duration, slower average speed and lower mean vertical distance [28]. A computer vision technique can record the stride length, back arch and swing duration, which are suitable for detecting cow lameness [50]. Furthermore, the gold standard in lameness detection is usually manual locomotion scores or, sometimes, claw inspections [51]. After image acquisition, the detection algorithm more closely matches the methods of locomotion scores. These methods have been widely studied in an attempt to capitalize on computer vision to detect cow lameness. The computer vision lameness detection system uses the traditional two-dimensional (2D) cameras, three-dimensional (3D) and thermal infrared cameras. Detection systems of different types of cameras had different shooting angles, data acquisition and detection methods and advantages and disadvantages. We will discuss them separately in this paper. When lame cows walk, they have a lot of different characteristics from those of healthy cows. Different studies have chosen different characteristics (such as uneven gait and back arch) [52,53] or built different lameness detection methods based on the same characteristics [49,53] to try to improve the effectiveness of lameness detection systems, which was the primary focus of this review paper. However, there are still some problems in the computer vision lameness detection system, which has caused the system to not be widely used. This paper will also introduce these issues, and we provide a discussion about some suggestions, as well as possible directions for future research.

The purpose of this paper was to review the development and research of the computer vision-based lameness detection technique for dairy cows, providing references and suggestions for future research approaches, as well as the development of computer vision-based lameness automatic detection systems, by analyzing the existing methods and technologies used in lameness detection. This paper is divided into four parts: In part 1, the main studies used in lameness detection (based on 2D computer vision) are reviewed and introduced, including the image processing, feature selection, classification and evaluation methods. Part 2 introduces the lameness detection techniques based on newer types of cameras, including 3D and thermal infrared cameras, as well as their advantages and disadvantages and a comparison with the 2D detection methods. Part 3 briefly introduces other sensors for lameness detection used in the current research and analyzes the inspiration behind other detection technologies for the study of computer vision-based lameness detection. Finally, the findings of this review are presented in Part 4. Part 5 summarizes the article and discusses computer vision-based lameness detection from three aspects—namely, research methods, verification methods and practical applications—to provide ideas and references for future research on lameness detection.

## 2. Materials and Methods

The key words lameness, automated, lameness classification, lameness detection, computer vision, image processing, locomotion scoring, dairy, and cattle in various combinations were entered into Google Scholar (https://scholar.google.com/), Web of Knowledge (http://wokinfo.com/) and Science Direct (https://www.sciencedirect.com/). This review includes peer-reviewed articles (132 articles) published between January 1989 and September 2020. 

### 2.1. 2D Computer Vision Detection

Two-dimensional computer vision technique applications show a high potential for detecting dairy cow lameness [54]; thus, researchers have begun to explore how to obtain and analyze characteristics that indicate dairy cow lameness using 2D computer vision techniques. Table 1 summarizes the characteristics, technique methods and results of the abovementioned 2D computer vision-based lameness detection methods for dairy cows. The conventional method for acquiring videos of cows is to select a passage or corridor at a farm and induce a single cow to walk down it [55]. If the farm does not possess a suitable configuration, a passage needs to be constructed [52]. According to research on a 2D detection system of cow lameness [47,49,52,53,56], a system diagram showing the camera, the passing alley and the computer is shown in Figure 1. The camera is placed to one side of the passing alley to ensure that the acquired side view of the cow walking will be clear and complete. The computer needs to connect with the camera, which is mainly used for image processing and running a lameness detection algorithm.

A 2D computer vision technique was first applied in cow lameness detection research to collect a dataset of walking cow videos. The purpose of recording videos of walking dairy cows that professionals could observe repeatedly was to obtain more accurate detection results than were possible through on-site observations [62]. The results demonstrated that using a 2D computer vision technique in conjunction with a digital rating system was both effective and reliable for detecting lameness. Subsequently, dynamic information about the symmetry of dairy cow gaits was found by researchers that could be analyzed using multiple frequencies of the video frames [58]. Then, more characteristics, including the positions of the cows’ hooves, the step time and the back arch features, were manually marked in the video to detect lameness [54]. The above research shows that a 2D computer vision technique was feasible for detecting cow lameness. Some methods of gait assessment, such as the numerical rating system, were valid and reliable and highlighted the importance of the detection of lameness in cows using multiple characteristics, including the trackway overlap, hoof step time and spine arch. These characteristics were extracted manually from the images. However, videos contain many frames, and it is time-consuming and labor-intensive to perform manual marking. To solve this problem, an image processing technique can be applied to detect lame cows. One traditional image processing technique extracted the landing positions of the cow’s hooves from the video and demonstrated good performance [52]. This system attained a 94.8% mean correlation coefficient between the vision-based calculated hoof locations and the manual reference. However, the study did not describe the testing environment in detail, and differences in environment, lighting and the background will affect the accuracy of the object detection of the key features of lameness detection [63,64]. Nevertheless, the research also demonstrated the innovation of the detection method of lameness and the method of locating cattle hooves [52]. With these studies, the cow lameness detection techniques based on 2D computer vision began to develop toward automated solutions.

Many studies exist that focus on using image processing techniques to extract better cow lameness features from videos [53,61,65,66,67]. This is the fundamental technique for automatic detection. The main problem in such image processing is accurately extracting the walking cow from the complex background. The classical method is background subtraction [52], but due to background/foreground contrast homogeneity, external factors can easily interfere with the background; thus, simple background subtraction methods have difficulty segmenting cows [68]. Therefore, the method of exaggerating the differences between the background and foreground was proposed, which has a good effect when extracting a cow’s back posture [53]. In the follow-up work, the tracking algorithm was improved to avoid mutual influence between cows when the distance between them was small [53]. To detect changes in leg posture, the template matching method was adopted to track the cow’s body and determine the cow’s leg positions; then, the three-frame difference method was used to identify the moving cow’s legs [65]. As a cow walks, its legs move in a fan-shaped track after the cow’s hooves touch the ground. During this period, the cow’s hooves remain at rest, and there are time and spatial differences between steps. Based on this characteristic, the researchers proposed an algorithm that considered these time and spatial differences using the local shape mutation caused by the static space of the cow’s hoof images as a feature for extracting the positions of the cow’s hooves [61]. When analyzing the pixel distribution characteristics of each frame image in a video, the study found that the target cow pixels and background pixels can be described by a bimodal distribution, while the two normal distribution proportions can be used to distinguish background pixels from cow pixels [66]. All the above studies showed good detection effects on various parts of cows, but the premise is that the detection of backgrounds changes very little. In practical applications, the background will change with time and light, which will greatly affect the video capture quality of the camera and the accuracy of the algorithm. Therefore, a deep learning technique was applied to extract cow lameness features from videos, which meets the requirements for the high-precision detection of the key parts of cows in natural scenes [67]. Compared with traditional image processing methods, a deep learning technique can improve the feature extraction of cows (Figure 2). However, using the same processing system, the deep learning detection speed was slow. Ensuring that the deep learning image processing algorithm works fast, such as by adding GPU processors to the system, will greatly increase the cost of the system. Therefore, how to find a balance between the detection precision and system cost is a problem associated with practical application [69].

After obtaining the motion information in the video image, the parameters related to lameness were selected and extracted for detection. These mainly involved the motion characteristics of cows as described in the five-point scoring system, including the back arch [49,53,54,59], head bob [49,53,62] and uneven gait [52,70,71]. In addition to using the characteristics of cows in the scoring system to detect lameness, researchers also found that correlations existed between some important variables and lameness by using an optical camera. These variables are also suitable for lameness detection, and they include the hoof placement times [54]; landing differences between the hind hoof and the front hoof [52]; the range and angle of movement of the front hoof of the cow [47]; the slope of the line connecting the head, neck and back of the cow [59] and the differences in the supporting phases [56]. To provide readers an intuitive understanding, this paper selected several images of cows from our lab’s experimental videos that contained different characteristics, including arched and nonarched cows and whether the hind hoof reached the position of the fore hoof (Figure 3). All the above-mentioned characteristics of cow movement are used to detect lameness, as there are differences in these characteristics between healthy and lame cows. However, in judging the relationship between different features and the lameness of cows, the difficulty of extracting different features from video images should also be considered. For example, in video images, the cow’s back is easier to extract in contrast to the joint angle. Therefore, in the detection system of dairy cow lameness, detection feature selection should enable easy identification of the feature being extracted, with high precision and a high correlation with lameness. Previous studies mostly used a single indicator to evaluate lameness. Single-indicator information collection is fast and makes the classification algorithm simple, but it cannot accurately and comprehensively characterize cow lameness [72]. For example, the back arch is used to detect cows; however, some cows that are lame do not show an arched back, while some nonlame cows show an arched back [59], including those with abdominal pain, pericardial disease and pleuropneumonia [73]. Additionally, the variables should not be affected by management or the environment. The criteria for the detection of cow lameness analyzed using the computer vision technique indicated that the method should include uneven gait and back arch [72]. According to a weight survey of various abnormal indexes of lame cow behaviors, the weights of general symmetry, tracking, spine curvature, head bobbing, speed, abduction and adduction were 24%, 20%, 19%, 15%, 12% and 9%, respectively [26]. Different indexes can represent different degrees of lameness, and each index had a relatively low weight. Using multiple features for lameness detection can make the detection more comprehensive [50,54]. Therefore, it is necessary to conduct a comprehensive evaluation of multiple indexes to detect lameness.

After acquiring feature information from the dairy cattle, it is necessary to classify lameness according to various characteristics through statistical analysis of the data, such as linear correlation analysis or spearman rank correlation analysis [54,58]. The conventional classification algorithm was first used to calculate the characteristic values using key positions of the cows, such as shoulder and hoof, as shown in the example in Formula 1 for the trackway overlap (Δ), which was defined as the distance of the hind hoof on the fore hoof position and was calculated by subtracting the position of the fore hoof from the hind hoof in the walking direction [52]. Then, the characteristic thresholds were defined according to the characteristic data of different lameness classes; for example, in some studies [53,59], the back arch curvatures of cows were used to define the thresholds. According to the different levels of lameness with different back arch curvatures, the thresholds of the back arch curvatures were set. However, because cows are complex individuals and time-varying living organisms [49,74,75], the variables used for detecting lameness are limited by the substantial differences between individual cows [76], and such differences must be considered by an effective automatic lameness detection system [53]. Previous studies defined thresholds and standards for lameness detection for different groups of cows but did not focus on individual differences [49,53,70]. Therefore, Viazzi et al. [49] proposed training decision trees to adjust the thresholds according to a cow’s body movement patterns to consider the individual differences in lameness detection. In that research, a cow walking video was divided into two datasets, and the population threshold and individual threshold were used to detect lameness. A BMP algorithm was used to monitor the lameness. The classification using population thresholds did not perform well, only correctly classified 76% of cow instances. The high variability between individual cows made it difficult to discriminate between contiguous classes. The use of an individual approach can compensate for this inaccuracy by considering the high variability among cows. The classification rate of individual thresholds was high (more than 85%) in all three lameness classes. Later, researchers not only began to use decision trees [50] but, also, k-nearest neighbor [60], a single-stream long-term optical flow convolution network [77] and other machine learning and deep learning algorithms to classify cow lameness, with accuracies that all exceeded 90%. The main process of detection (2D) is shown in Figure 4. A consensus has yet to emerge regarding which was the most appropriate with a wide range of classifiers (e.g., thresholds, support vector machines and logistic regression) [48]. However, when multiple features are used to detect lameness, a more accurate detection result can be obtained based on many features. In follow-up research, more features and factors will be considered, and more accurate results will be required, so machine learning algorithms will have a broader application space.
(1)$$\Delta_{\mathrm{Left}}=X_{\mathrm{FL}}-X_{\mathrm{HL}}$$
$$\Delta_{\mathrm{Right}}=X_{\mathrm{FR}}-X_{\mathrm{HR}}$$
where $X_{\mathrm{FL}}$ denotes the step position of the left front hoof,$X_{\mathrm{HL}}$ denotes the step position of the left hind hoof, $X_{\mathrm{FR}}$ denotes the step position of the right front hoof,$X_{\mathrm{HR}}$ denotes the step position of the right hind hoof, $\Delta_{\mathrm{Left}}$ denotes the trackway overlap of the left body side and $\Delta_{\mathrm{Right}}$ denotes the trackway overlap of the right body side.

For the evaluation of the lameness prediction model, specificity was the proportion of nonlame cows that were correctly detected as nonlame. Conversely, sensitivity was the proportion of lame cows that were correctly detected as lame. It was suggested that a greater than 90% sensitivity and 99% specificity would be of significant value on most farms and could satisfy farmers [48]. However, the existing research algorithms could not meet this standard, so the performances of computer vision lameness detection systems need to be improved for them to be applied. 

### 2.2. New Cameras

In studies that address cow lameness detection by computer vision, the new camera types mainly include 3D cameras and thermal infrared cameras. Compared with a traditional 2D camera, the advantage of these new cameras is that they can obtain more information, such as image depth and temperature information [24,78]. This information not only improves the accuracy of cow lameness detection but can also include physiological indicators that cannot be extracted using 2D computer vision [79]. Table 2 summarizes the characteristics, methods and results of some of the typical lameness detection methods for dairy cattle that use 3D and thermal infrared cameras.

#### 2.2.1. 3 D Computer Vision Detection

There are still some limitations in the computer vision research involving 2D cameras applied to cow lameness detection that make the development of a fully automatic detection system impossible. First, many farms have inadequate space, and not all farms have good places to install side-view cameras [55]. Second, changes in lighting conditions cause noise and shadows in the images that affect the feature extraction process [67]. Finally, background changes make it difficult to segment the cow from the image [67,79]. In contrast, 3D cameras can avoid the above problems to a certain extent [78]. Viazzi et al. [79] analyzed the feasibility of using 3D cameras to detect cow lameness compared with 2D cameras. The 3D camera was installed above a corridor entrance door, and the curvature of the cow’s back was selected as the detection method. The accuracy of these two results reached more than 90%, and the specificity of the 2D and 3D camera algorithms was 91% and 95%, respectively, while the sensitivity of the 2D and 3D camera algorithms was 76% and 82%, respectively. Overall, 3D cameras can be used for detecting cow lameness and can achieve good performances. This is a very meaningful finding. From the aspect of feature acquisition, 3D cameras can reveal the back postures of cows more accurately [78]. From the aspect of equipment installation, 3D cameras were installed above the channel and aimed down, which enabled a top view of walking cows. Compared with 2D cameras, which need to be placed several meters on one side of the channel, 3D cameras provide more space saving and are suitable for more farms [79]. Although 3D cameras also have some limitations, such as their sensitivity to natural light, small field of vision and inability to analyze gait information, they still have high research value for developing automatic detection systems to classify cow lameness. The main process of detection (3D) is shown in Figure 5. A study on a farm [86] in which cow behavior sensors and milk yields and milk quality sensors were already present to detect lameness showed that the 3D video-based system outperformed the behavior and performance sensing techniques previously applied on this farm for lameness detection, as well as demonstrated that it is worthwhile to consider regardless of whether there are other sensors available [86].

Some researchers optimized the developed algorithm to reduce false positives (such as a cow tripping or slipping) and optimized the classification performance by multiple consecutive measurements using 3D video [84]. Four continuous locomotion score average values were used as classifier inputs, and three different classification models were tested. The results revealed that continuous measurements improve the classification accuracy of the system compared with independent observation results [84]. It was common for cows to slip when they were walking, especially when the ground was slippery or a farmer was driving them, which had a great impact on the detection of lameness. As mentioned above, researchers often artificially select videos of cows walking normally for algorithm research, but this situation cannot be guaranteed in practical applications. The authors observed that cows slipped and stood still for a short period of time. Continuous measurements could be used to reduce false alarms, but a better method would be to eliminate such anomalous data, which would require a posture judgment process, including leg posture, which should be easier in 2D than in 3D, because the former collects side views of a cow walking, whereas 3D cameras only collect images of a cow’s back. Additionally, since the view was from the top, the previous study mainly judged the lameness based on the back arch, which is not completely reliable, because some lame cows do not have a back arch, and, similarly, some healthy cows show a back arch phenomenon [59]. Some researchers have found that not only the back arch information but, also, the speed can be calculated from 3D videos [86], the curvature angle of the back around the hip joints [89] and the movement information regarding the hind legs and spines [85] of dairy cows, which can be used to detect lameness. Moreover, the curvature of the spinal column in 3D images can be used not only to judge the severity of the lameness but, also, to distinguish left- or right-side lameness based on the curvature direction [88]. In general, the detection of lameness in cows using 3D cameras is feasible and effective. Compared with 2D cameras, 3D cameras have lower requirements for environmental conditions and can obtain images more easily. In addition, 3D cameras obtain more comprehensive back information and are more suitable for long-term observation and data collection for statistical analysis [76]. However, due to the larger amount of 3D image data, the computational load of the computer is heavier, and the camera has a smaller field of vision [79]; thus, there is a need to control the cow traffic [89]. The gait is important in the detection of lameness in cows [48], and the inability to obtain leg images limits the comprehensiveness of the 3D camera’s detection of lameness in cows.

#### 2.2.2. Thermal Infrared Camera

Infrared thermography (IRT) is a noninvasive, non-radiation, rapidly evolving diagnostic method, and it can measure the surface temperature of an object [90]. Since hoof inflammation can cause the skin surface temperature to rise [91,92,93], many studies detect the lameness of cows by the change in hoof temperature [94,95,96]. Before IRT was used to detect hoof disease in dairy cows, a large amount of data was needed to train a model to learn the relationship between temperature and hoof disease and to determine the non-hoof disease interference factors that affect the hoof temperature of dairy cows [80]. When a cow’s hoof is damaged, the surface temperature increases [91,95]; thus, IRT can detect lameness in a cow’s hind leg by focusing on the coronary artery area of the cow hoof [97,98]. Studies have found no significant difference in temperature between different hoof diseases, but they did find a significant difference in hoof temperature between diseased and healthy cows [82]. Temperatures on a farm change with the season and over time, which can affect the thermal infrared detection results [99], and individual animal variabilities combined with thermoregulation increase the detection difficulty [24]. Therefore, some researchers proposed using the difference in temperature between the left and right feet of each cow to detect hoof diseases to eliminate the influence of external factors and individual specificity [82], but the adopted threshold value also classifies many undamaged hoofs as diseased hoofs. Therefore, the selection of the threshold value still needs further study. Due to the different temperatures resulting from different hoof positions, the image with the highest temperature at the heel of the hoof is the most accurate representation of diseased hooves in detecting lameness, and the sum of the highest temperatures of the coronary artery band and the skin can be used as the judgment basis [83]. In addition to the maximum temperature, the 95th percentile and the standard deviation can also be used to distinguish lameness, especially when urine or feces are present in the image [87]. Although IRT is considered to belong to the computer vision field, its main use with respect to this study is to detect cow hoof temperatures during lameness detection processes. Infrared thermography has great potential as a diagnostic method for lameness in dairy, especially for early detection and prevention, and it can compensate for some of the deficiencies in 2D or 3D lameness detection, because early lameness motor characteristics may not be as significant. Thus, it is best used in combination with other diagnostic imaging methods [93]. During detection, only the temperature is used to determine the parameters, and most of the image information is used for temperature positioning. Therefore, the image information of IRT can be considered for more applications in lameness detection.

### 2.3. Other Sensors

Other research on the detection of lameness in dairy cattle has focused mainly on changes in movement [100], changes in the load-bearing distribution of the legs [101] and natural behavioral changes caused by lameness [15,102]. These methods use a contact sensor to collect movement or pressure information to detect lameness. Although the methods differ from those of computer vision, the main research ideas provide instruction and guidance for research on cow lameness detection based on computer vision. This section briefly introduces these other lameness detection methods. Contact-type lameness detection sensors mainly include pressure sensors and accelerometers. In the study of using pressure sensors to detect lameness, a weighing platform [103,104,105,106,107] or a pressure-sensitive mat [108,109,110,111,112,113,114] was placed where dairy cattle were walking or standing. When a cow’s hoof contacted the sensor, the researchers obtained information on both the location of the cow’s hoof and the pressure exerted on the underlying surface. This information was converted into load-bearing asymmetry or multiple gait asymmetry variables, including “length”, “time” and “overlap”; then, a judgement was made regarding whether the cow exhibited lameness. Lameness can also affect dairy cows’ daily activities and cause behavioral changes [115]. Accelerometers can be used to record dairy cow behaviors [48,116,117,118,119,120], activities [97,121] and lying-down times [122]. These data are then analyzed to determine whether the activities are abnormal, to indirectly detect dairy cow lameness and to predict lameness via long-term data comparative analyses.

Compared with computer vision detection methods, contact sensors have several advantages, including easier data acquisition, stronger mapping relationships between the data and lameness characteristics and simpler research and decision-making methods. These studies also reveal some deficiencies in the computer vision-based techniques for detecting cow lameness. One of the most important deficiencies is that a computer vision technique cannot accurately obtain the characteristic information of cows through image processing, and errors in image processing lead to errors in characteristic information and then affect the judgment involved in the decision-making of lameness detection. However, due to the noncontact information acquisition method and the detection method being similar to the locomotion score system, this technique is worthy of being studied. Therefore, in the next chapter, we discuss the development of a cow lameness-detection technique based on computer vision.

## 3. Discussion

Lameness remains an important problem for dairy farming. Due to the great difficulty and cost of the manual detection of lameness, it is necessary to accurately and automatically detect the lameness of cows. The main advantages of lameness detection in cows based on computer vision methods are that these methods are cheap and noncontact [65]; however, there are still some problems and obstacles related to the research and application of these methods. In this paper, the problems and developmental prospects of computer vision lameness detection in dairy cows are discussed in terms of three aspects: detection methods, verification methods and application implementation.

### 3.1. Detection Methods

Locomotion scores and foot pathologies are usually used as the reference standards for the validation of automatic lameness detection systems (ALDSs) [28]. The current research on lameness detection using computer vision is largely based on manual detection and scoring systems, such as the detection of a cow’s back arch or uneven gait. However, there are differences between manual and computer vision detection. A computer vision technique can maintain a perspective for a long time, store the data and extract the slight differences of the features. Manual detection can adapt to and shield from environmental changes and ignore unnatural cow gaits (e.g., slipping and stopping). Therefore, the application of artificial detection methods do not fully exploit the native advantages of computer vision. In this paper, three suggestions are proposed for research methods of lameness detection in cows by computer vision. (1) Explore the new methods that are more suitable for computer vision. Through the exceptional data collection abilities of cameras, we suggest the exploration of new tests in follow-up studies that can solve the existing problems, such as the influence of individual specificity among dairy cows and the detection of lameness in hooves. We also hope that the computer vision system can accurately obtain the characteristic data and can be more suitable for use with computer algorithms to detect lameness. Although the principles of the computer vision lameness detection method are similar to those of the manual observation detection method, it should not be limited by the manual detection method in the computer vision lameness detection. New features and methods more suitable for computer vision lameness detection should make the system perform better. (2) Explore the methods that apply multi-feature fusion to detect cow lameness; these approaches make the detection more comprehensive. Due to the individual specificity among cows, the lameness manifestations and degrees of each lameness cow are different, and single characteristics are prone to causing misdetections and missing lameness detections. Complementary methods can improve the accuracy of lameness detection and classification, reduce interference and improve the detection robustness. (3) Reducing the number of inputs to the decision algorithm. Overall, there is a connection between using multiple lame cow walking features acquired during the movement process. When analyzing the coupling relationships between these features, it is better to use fewer features that represent more feature input of the decision algorithms and reduce their calculation burdens. Based on such reductions, we can more fully understand the essence of the movements of lame dairy cows and fully capitalize on the advantages of computer vision. Moreover, the deep learning technique brings a new direction for the detection of lameness in cows with computer vision. The convolutional neural network performs well in target tracking and feature extraction in image processing, and it has been proven to be able to accurately extract the key location information of dairy cow lameness detection [56,67,123]. However, the spatiotemporal changes of moving objects in the video can also be well-detected by a deep learning technique [124,125,126]. The dynamic information of an uneven gait caused by the lameness of cows can be accurately distinguished by a deep learning technique from walking videos of cows [77]. In general, we suggest that the possible future technique solution is to use a deep learning technique to extract the cow lameness characteristic information from a video. Multiple lameness characteristics should be used as the basis for detection; an in-depth analysis of the relationships between each lameness characteristic should be performed to establish lameness detection algorithms, or large amounts of data using machine learning should be used to determine a lameness characteristic and lameness class mapping relationship.

### 3.2. Verification Methods

Today’s lameness detection research typically uses professional observations as the validation data for research results; however, lameness is a process, rather than a binary characteristic (e.g., a simple 0 or 1) [30,31]. Thus, no clear boundary exists between the varying degrees of lameness. This approach introduces subjective factors that may not be appropriate, which could have an impact on the conclusion of the study. Consequently, improving the reliability of lameness detection verification is of great significance for the accuracy of the research and the realization of the results. Through previous studies, we have learned that, compared with all detection methods, a thermal infrared camera is able to detect mild lameness earlier [127], and it has a higher detection sensitivity, but these methods require taking close-up images of cows’ hooves. Innovative technological methods that establish detection systems using only thermal infrared cameras are still lacking. Nevertheless, thermal infrared video techniques provide a good option to be used for verification [93]. Due to the error rates of IRT methods, their results should still be combined with professional observations. If necessary, studies can also refer to the pressure, activity and other parameters to ensure that the results of the verification set are accurate and reliable. With the development of machine learning, the concept of datasets has become popular, and various open datasets provide convenience for researchers worldwide. In computer vision research on the detection of cow lameness, it would be helpful to construct a video dataset of cow lameness. The cows in the dataset should be classified according to the unified standards recognized by most researchers, including the large number of different lameness grades and types of cows. This dataset would help promote research on cow lameness detection. Researchers could directly use this dataset for image processing, algorithm analysis and verification of the proposed method. Of course, establishing a dataset is not simple; it requires substantial work to collect data suitable for different research methods. The results of lameness grading should not only be calibrated by one person or by a research institution but should also be analyzed and calibrated by multiple researchers involved in computer vision-based cow lameness detection to obtain widely recognized results.

### 3.3. Application Implementation

Although the methods of cow lameness detection based on computer vision have been studied for many years, they have not been widely applied. An automatic lameness detection system must be effective, reliable and feasible [28]. Based on the problems mentioned in the manuscript, this part discusses why cow lameness detection systems based on computer vision have not been widely adopted and how such adoption can be increased.

The result of the survey “Why are automatic cow lameness detection systems not popular?” shows that “before automatic detection cow lameness systems can be applied in practice, low-cost systems with high-detection performance must be available” [69]. The cost and detection ability of such systems are the two main factors that affect their adoption in real-world lameness detection.

Compared with manual detection, despite that the automatic detection system has equipment costs, the returns from automated lameness detection may still be worthwhile [128]. However, lower costs would be more acceptable [69]. A computer vision system is composed primarily of a camera and a computer, such as an industrial control computer. Note that we do not discuss the computer itself in this study. Cameras (2D cameras, 3D cameras and thermal infrared cameras) have wide price ranges, and it is undeniable that data collected by inexpensive cameras will cause problems in the subsequent processing (such as ghost images) that affect the test results. The purpose of this paper was to summarize and discuss dairy cow lameness detection using computer vision; thus, it is unable to provide suggestions regarding equipment costs. However, with the development of electronic techniques, the price of electronic equipment at a given performance level tends to decrease (e.g., mobile phones). Moreover, a computer vision technique can not only be used in cow lameness detection but is also useful in research that focuses on detecting physiological information or diseases, such as body measurements and mastitis detection [78,129]. When many studies are integrated into a computer vision system, the equipment costs of various detection functions are reduced in an indirect manner, so the problem of equipment costs will gradually diminish as the technique develops [78,129]. Computer vision-based cow lameness detection requires a specific detection environment [55], and there are stricter requirements for standardization and automation on farms, which is one reason why automatic cow lameness detection systems are not popular. Usually, the systems require that the cow must pass along a defined channel [52] and that there should be space beside or above the channel to place the system equipment. Computer vision-based detection techniques have become suitable for farms only as channels have become common and the space for the system equipment is sufficient. The cost of creating a channel solely for lameness detection can be avoided [52].

Regarding the detection ability of cow lameness systems based on computer vision, farm staffs are primarily concerned with the false-positive percentage, the missed lameness percentage and the accuracy of the indication regarding which leg exhibits lameness [69]. Another reason why automatic cow lameness detection systems are not popular is that the results fail to meet the requirements of farm workers [48]. As mentioned above, using multiple features to detect lameness effectively makes the detection more comprehensive. In the current research, 3D cameras mainly use the image information from above and behind the cow to detect lameness; this approach cannot help in analyzing cow hooves, and the sheer size of the 3D image data is a challenge for the detection speed. Although thermal infrared cameras can detect lameness earlier based on detecting cattle hoof temperatures, these systems require a close-up camera shot, which is inconvenient to acquire. A 2D camera can acquire a side view of a walking cow, which carries more characteristic lameness information and can be used for the detection of a cow’s hooves [61], but its installation is difficult [79]. Therefore, that question was worthy of further study. According to our point of view, there was another problem that has made the automated cattle lameness detection system unpopular. All computer vision methods used to detect lameness in cows are based on their natural walking. In the studies, the data used were manually selected. In practical applications, cows sometimes slide or stop during the walking process, which affects the detection results. Therefore, in practical application systems, it is very important to distinguish and exclude unnatural walking.

In addition to the system costs and detection ability, some additional problems affect the deployment of these systems. For example, 2D camera images are disturbed by illumination conditions and background changes and have strict requirements for placement and positioning. When long-term and continuous measurements are needed [76], environmental factors (e.g., dust and water vapor) may adversely affect the equipment hardware. These problems need to be solved by further research, algorithm improvements and new hardware devices. Therefore, although computer vision-based lameness detection methods for dairy cows have been studied and improved for many years, we believe that they still need to be developed further to establish a better application system. It is necessary to identify and address the influence of various nonideal conditions (e.g., the cows paused and overlapped) and environments on the detection results in practice and improve the robustness of the detection system, which was the key to the application of the computer vision dairy cow lameness detection system.

## 4. Conclusions

Computer vision-based technologies used for cow lameness detection have the advantages of a noncontact application and moderate prices, which can effectively classify lameness according to the walking status of dairy cows to improve the welfare of dairy cows and reduce economic losses. This paper aimed to provide an overview of the results in this field of research. It classified these techniques according to the types of cameras used in the computer vision systems, including 2D, 3D and thermal infrared cameras. Since different types of cameras can provide different images of cows, the research methods are also different. We separately reviewed their key steps in lameness detection, including feature selection, the feature acquisition technique and lameness detection method, since most research was focused on these steps. At the same time, this paper compared and discussed the advantages and disadvantages of the different types of cameras used in lameness detection systems. In the study of a computer vision detection technique for dairy cows, there were still some problems and obstacles in the detection effects and application. This paper discussed the problems and development prospects of this technique from three aspects: detection methods, verification methods and application implementation. Development aimed at the application and popularization of the computer vision lameness detection system for dairy cows still faces technical problems and limitations related to its application on farms, which requires further exploration and research.

## Figures and Tables

**Figure 1 sensors-21-00753-f001:**
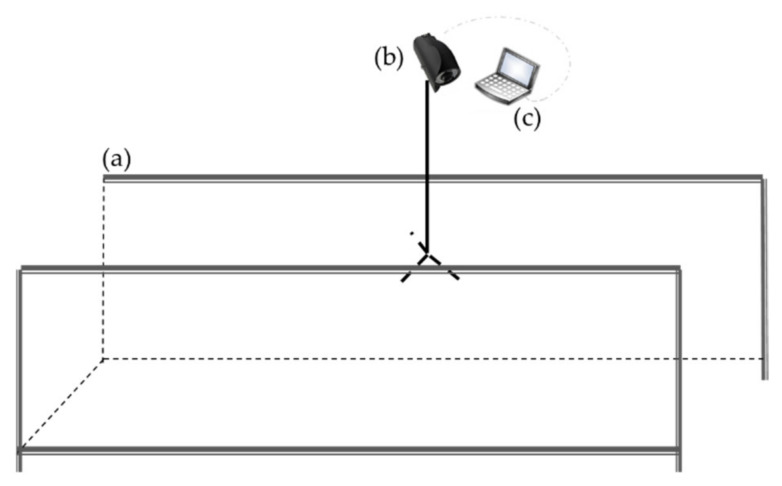
Top view of the cow lameness detection system based on a 2D camera: (**a**) the passing alley, (**b**) the 2D camera and (**c**) a computer or industrial computer.

**Figure 2 sensors-21-00753-f002:**
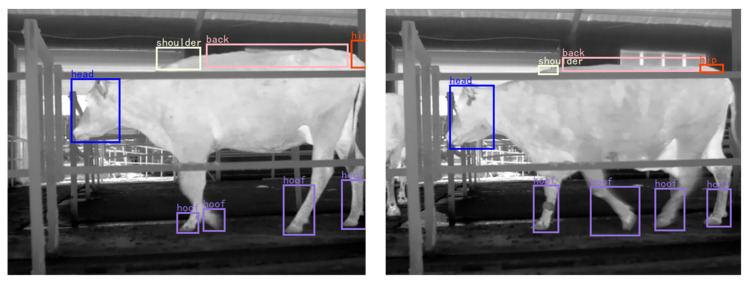
Examples of the feature extraction of cows using a deep learning technique.

**Figure 3 sensors-21-00753-f003:**
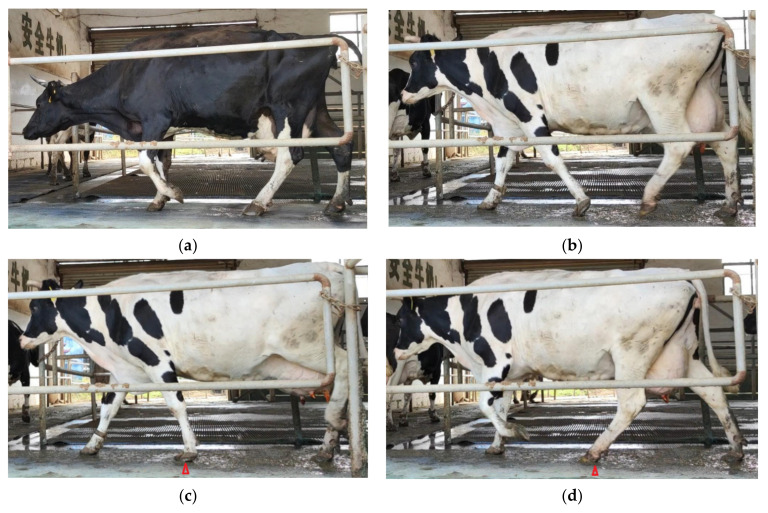
Examples of cows with various characteristics: (**a**) a cow with a back arch, (**b**) a cow without a back arch, (**c**,**d**) the hind hoof reached the front hoof position and (**e**,**f**) the hind hoof did not reach the front hoof position (Δ represents the front foot landing position).

**Figure 4 sensors-21-00753-f004:**
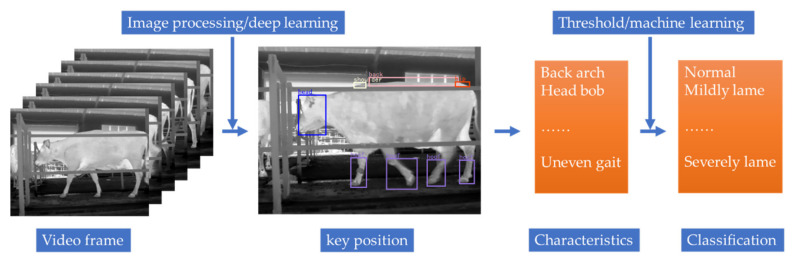
The process of detection (2D).

**Figure 5 sensors-21-00753-f005:**
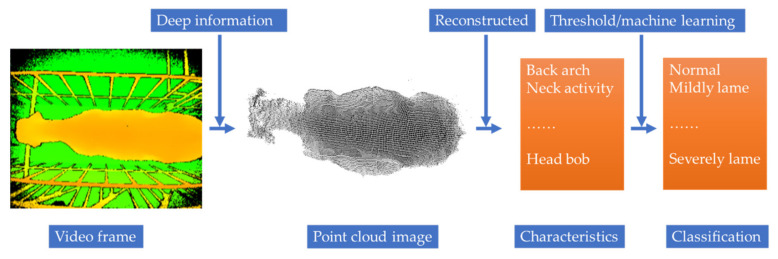
The detection process (3D).

**Table 1 sensors-21-00753-t001:** Cow lameness detection methods based on 2D computer vision.

Reference	Camera	Test Environment	Objective of the Study	Characteristic	Research Technique Method	Algorithm Used	Result
Flower and Weary [57]	Panasonic AG-195 MP	A rope barrier in the test alley	Explore how hoof pathologies affect dairy cattle gait	Back arch, head bob, tracking-up, joint flexion, asymmetric gait, and reluctance to bear weight	Observation by a trained observer	/	Evaluation methods such as numerical rating systems are effective
Bahr et al. [58]	Canon Powershot A620 zoom camera	Center of the corridor	Explore the possibility of capturing hoof locomotion with image parameters and calculate the relationship between the parameters and lameness	Hoof step time	Manual marking and difference calculation	Spearman rank correlation coefficient	Hoof step timecan be used to detect lameness; the correlation is 84%
Song et al. [52]	AVT Marlin F-131C	A 6-m long, 0.9-m wide passing alley	Demonstrate the possibility of capturing hoof locations of cows by vision and assessing the correlation between automatically calculated hoof trackway and visual locomotion scores	Trackway overlap	Image processing, correlation analysis	/	The accuracy of locating trackway overlap locations from images is 94.8%; trackway overlaps have a positive linear relationship with lameness
Bahr et al. [54]	Canon Powershot A620 zoom camera	A corridor walk from the barn to the pasture ground	Develop and analyze image parameters correlated with expert gait scores that were applicable for lameness detection	Trackway Overlap, Hoof step time, Back Arch	Manual marking correlation analysis	/	Three characteristics are positively correlated with the degree of lameness
Poursaberi et al. [53]	Canon Powershot A620 zoom camera	A 1.2-m-wide and 6-m-long concrete corridor	Attempts for automatic vision-based lameness detection based on back posture analysis	Back arch	Hierarchical background/foreground exaggeration, three-point curvature calculation	New multi-filtering scheme	Sensitivity of 100%, specificity of 97.6% and accuracy of 94.7%
Poursaberi et al. [59]	/	The corridor after the milking parlor	Propose a method for real-time lameness detection based on back posture analysis	Combined analysis of back posture and head position	Image processing, calculating the curvature of a double ellipse	Body Movement Pattern (BMP) detection algorithm	97.4% correct rate of classification
Pluk et al. [47]	Guppy F-080C camera, SV-03514 3.5-mm lens	A 1-m-wide and 6-m-long rigid bridge with a separation gate	Describe a synchronized measurement system combining image and pressure data for lameness detection	Limb change angle	Timing and positioning of a pressure-sensitive mat, image processing	Kruskal-Wallis and Wilcoxon rank-sum tests	The range and angle of forefoot movement are important variables in lameness classification
Viazzi et al. [49]	Canon 17-85 IS USMlens	A 4-m-wide, 7-m-long corridor with a concrete floor	Develop and test an individualized version of the body movement pattern score	Back arch	Decision tree classification	BMP detection algorithm	91% true positive rate, 6% false positive rate and accuracy of 91%
Zhao [50]	integrated web camera Hikvision Inc	A 2-m-wide alley with a solid concrete floor	Analyze leg swing using computer vision techniques	Leg swing	Decision tree classification	Leg swing detection algorithm	Sensitivity of 90.25%, specificity of 94.74% and accuracy of 90.18%
Song et al. [60]	/	/	Explore the possibility of lameness detection using the fitting line slope feature of head and neck outline	Fitting line slope feature of head and neck outline	Comparative study	Normal background statistical model of local circulation center compensation in track-distilling data of KNN (K-Nearest Neighbor)	Accuracy of 93%
Kang et al. [61]	/	The corridor before the milking parlor	Computer vision technique for cow hoof positioning	Spatial-temporal characteristic of hoofs	Image processing	Spatiotemporal difference algorithm	The positioning error of cattle hooves is less than 11. 3 pixels ^1^

^1^ The “/” means there was no discussion of the factor in the article.

**Table 2 sensors-21-00753-t002:** Detection method of cow lameness based on 3D and thermal infrared cameras.

Source	Camera	Test Environment	Objective of the Study	Characteristic	Research Method	Algorithm Used	Result
Nikkhah et al. [80]	FLIR Inframetric 760, Boston, MA	Within the barn	Explore the relationship between hoof temperature and hoof health of cows	Temperatures of cow hooves	Chi-square analysis	/	Using infrared thermography (IRT) to measure skin temperature may reveal inflammation associated with laminitis in the early/middle stage
Alsaaod and Büscher [81]	Longwave thermal camera	Milking parlor	Investigate IRT as a noninvasive diagnostic tool for early detection of foot pathologies in dairy cows	Temperatures of cows’ hooves	Analysis of temperature difference between healthy and diseased hooves	Threshold classification	The sensitivity of thermal infrared imaging to detect hoof damage was greater than 80%
Stokes et al. [82]	/	Milking parlor	Examine the potential of IRT as a noninvasive tool for rapidly screening for the presence of digital dermatitis	Temperatures of cow hooves	Comparison of temperature changes in cow hooves caused by different hoof diseases	Threshold classification	Damage to hooves and skin causes a rise in peak skin temperature
Alsaaod et al. [83]	Ti25 Thermal Imager	In a closed, indoor environment	Evaluate IRT as a tool for the detection of digital dermatitis lesions and to determine an optimal temperature cut-off value	Temperatures of cow hooves	The two highest temperatures were used to evaluate disease in hind feet and hooves	Threshold classification	The sensitivity of hind foot disease detection was 89.1%, and the specificity was 66.6%
Viazzi et al. [79]	3D Kinect camera 2D Nikon D7000 camera	The alley after a sorting gate	Evaluate the use of a 3D camera from the top view to improve the back-posture extraction and to compare it with the 2D camera	Back arch	Decision tree	BMP detection algorithm, 3D back posture calculation algorithm	A 3D camera method is suitable for an automatic lameness detection system
Van Hertem et al. [84]	Microsoft Kinect Xbox 3D- camera	After-milking sorting gate	Optimize the classification output of a computer vision-based algorithm for automated lameness scoring	Back arch	Classification models such as logistic regression of ordered polynomials	BMP detection algorithm	Continuous measurements of cow lameness can improve the classification ability of a computer vision system
Jabbar et al. [85]	/	A custom race next to the milking parlor	Examine the ability of the spine arch analysis method to detect early-stage lameness	Spinal posture and gait	Image processing, data feature analysis	Threshold classification	Accuracy of lameness detection was 95.7%
Van Hertem et al. [86]	Kinect	Corridor	Evaluate whether a multi-sensor system was a better classifier for lameness than the single-sensor-based detection models	Back arch and speed	Comparison between single predictor and multivariate analysis	Binary logistic regression	Gait and posture measurement systems based on video are superior to the behavior and performance sensing technique for lameness detection
Harris-Bridge et al. [87]	FLIR SC620 camera	claw trimming crush	Determine whether the temperature data were more effective and accurate in detecting lameness	Temperature of cow hooves	Scatter plots and Pearson’s Product Moment correlations	Parametric statistical, linear model, maximum temperature detection	The highest temperature is the most accurate measurement method
Hansen et al. [88]	3D Kinect-like depth camera	a narrow walkway beneath	Explore a methodology for simultaneously monitoring multiple animal health parameters	Curvature of the spinal column	Image processing, spatial analysis	curvature of the spine threshold classification	Accuracy of lameness detection was 83% ^2^

^2^ The “/” means there was no discussion of the factor in the article.

## Data Availability

No new data were created or analyzed in this study. Data sharing is not applicable to this article.

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
