# Peer review of "A Review: Development of Computer Vision-Based Lameness Detection for Dairy Cows and Discussion of the Practical Applications"

_sensors, 2021, doi:10.3390/s21030753_

Round 1

Reviewer 1 Report

No relevant comments to do. The only suggestion could be about the conclusion, that can seems more a summary that a real conclusion of the paper.

I would have expected something more about possible future technological solutions.

Reviewer 2 Report

The manuscript is an interesting approach of one of the technologies that is available to develop tools to detect different health issues, as lameness. The authors divide the manuscript in sections to explain what has been done and the different technologies that are currently available.

General comments:

Even though the paper addresses a hot topic. I considered that is missing some important aspects as the different algorithms used to the detection of lameness. Is not only a matter of a camara or a tool is also how the data that is generated is analyzed, and this is missing in the manuscript.

There are sections of the paper that are repetitive and there are some lines that somewhat contradict what was mentioned before, see lines 62, 73 and 77, for example.

In the introduction is not clear what does lines 44 to 57 contribute to the review. I will suggest rewriting some of the sections of the manuscript as the introduction: focus more on the computer vision, which is the main objective of the paper, also explain briefly the difference between the 2D and 3D. Also rewrite the Application Implementation section and the conclusions.

I also suggest adding information about the data analyses approach and therefore discus about it.

Section about detection methods is not clear, I suggest rewriting it. There are sections that seems copied from another study (see line 419), what study?

In general sentences are very long and make hard to follow the idea, therefore consider reviewing the manuscript to make it easy to follow.

Specific comments 

L16: Considerer changing methods for tools

L21: The manuscript does not discuss how to improve the accuracy and practicability- I suggest rewriting the abstract and stick to what was done and include some conclusion of the review.

L29: ‘the’ instead of ‘in’

L31: Add a reference

L34: ‘while’ instead of ‘and’

L34: Clarify what ‘in other cows mean’ (e.g., other breeds? If there are other breeds considerer adding them). Also clarify that this study was done in grazing systems, since the lameness incidence is not the same under different production systems.

L42: Consider changing fecundity with reproduction

L43-44: Consider rewording

L62: Is not clear

L70: What do you mean by accuracy of the score?

L72: Is really detected by farmers, or without the scoring system?

L76: Consider changing ‘the score’ with ‘and’

L72-78: Considerer rewriting, is repetitive and not clear.

L84: Add reference

L86: delete ‘as’

L87: Add a ‘.’ after [48].

L88: what does “mature research” mean?

L90: ‘was’ instead of “is” - Change the verbs to the past tense to all the manuscript

L92: Add ‘.’ After approaches

L94-107: Rewrite this section according to what you really did.

L102: Analyze, how?

L102: delete ‘the paper’

L113: did you put a filter based in date for the search- if yes clarify why. Also is not clear why the dates go until October 2020, but the final search occurred in July 2020?

L122: Cow farm or farm?

L126-131: Is this an example- is not clear

L126: delete ‘that’ and ‘need to’ instead of was

L135: Delete videos after walking

L137: Add reference

L144: Which characteristics? Give more context

L142: Something is missing

L145: Not clear

L147: What are good experimental results

L152: Add ‘.’ After [51]

L154: delete because

L155: explain what does complex background mean and delete ’in’

L157: Which algorithm? add reference

L159: Add references

L187: It is not clear- rewrite it

L190:  there is a ‘.’or ‘;’ extra and delete ’how to find’

L191: add reference

L196: change “proved’ with ‘find’

L201: needs more explanation

L218-219: Change therefore in one of the sentences

L221: what accuracy? There is no number reported before

L226: which is the characteristic threshold?

L227: are cows’ systems?

L230: Add reference

L231: I suggest expanding this section, as is the core of the manuscript

L234: ‘S’ should not be capitalized

L235: what are good results?

L237: Why is machine learning difficult to understand?

L242: Sensitivity is repeated-

L243: is this really true?

L245: This reference is not applicable here.

L246-259: review this section, is not clear what are you trying to say here

L254: better than what?

L279:4 and 6 percent lower: Considerer changing by the real values.

L280: 6 percent higher: Considerer changing by the real values.

L285:’installed above the channel’- Need more explanation

L291: Which techniques? This farm? More context and explanation are needed- Add reference

L293: Add references

L3299: Add references

L303: how can you modify the data?

L305: Is this true?  More context and explanation are needed

L309: It contradict what is stated in line 307.

L318: How does camaras collect statistics?

L325-331: Rewrite this part, is very difficult to follow

L332: Is this really the goal of the IRT? Or it was the objective of the study cited?

L339: Is repeated

L353: delete research

L379: What does intuitive data stands for?

L402: Is this accurate? Considerer rewording

L411: Is not clear the suggestion types? It is a subjective classification

L419: Which study?

L469: Where is this discussed on the manuscript?

L484: BCS is not an example of what is mentioned before

L504: This was already stated elsewhere in the manuscript

L510: Consider adding ‘.’ After system. Delete ‘including that’ and add ‘For example,’

L517: Give an example of a non-ideal condition

L519: Rewrite, is not clear

L528: this is not a conclusion

L530: delete has

L532: ‘machine learning properly’ meaning? There is not discussion of ML techniques in the manuscript

L533: What does give more consideration to farmers mean?

L539: Different font

Figure 1 is very similar, not to say identical to the one published on the JDS (https://doi.org/10.3168/jds.2020-18288) by the authors.  I suggest that you cite the paper and adapt or change the figure.

Table 1 and Table 2. I suggest adding a column with the objective of the study. Also, add another column with the statistical analysis or algorithm used. What does the “-“ mean in table 2. Be consistent in how the results are reported in the tables, for example true positive of Viazzi et al. are as “0.96” whereas Song et al. are as percentage. What does “/” mean in table 2. Define IRT in table 2. Change “judge” to “evaluate” on table 3

Reviewer 3 Report

This review shows the evolution of lameness detection through computer vision analysis .

The paper is clear and well written, references are updated with recent citations

Reviewer 4 Report

  • Good thorough search of literature, however a review paper should be more than this.
  • The paper remains a list of publications with tables summarising lots of methods and results. The paper needs to provide readers a clearer view of the fundamental technologies, their effectiveness and trends.
  • The author should pick out a few important methods and explain concisely to readers so that readers could choose the best techniques for their problems. For example, what are the best 5 or 10 techniques should I know? What are the most interesting results others have achieved so far?
  • A paper is about computer vision technology but has very long paragraphs, and very few figures or diagrams. Need more figures including those from cited papers. Could the authors summarise the literature in one graph? Could the author introduce one figure for each section?
  • Some examples of a review paper:

https://www.sciencedirect.com/science/article/pii/003132039390135J

https://link.springer.com/article/10.1007/s11042-020-09004-3 

This one from Sensors journal and I don't recommend because it has the same problems https://www.mdpi.com/1424-8220/19/5/1005/pdf 

Round 2

Reviewer 2 Report

Even though the manuscript has improved, there are still long sentences that make difficult to follow the idea. Likewise, you still have some sentences that a repetitive, for example in the abstract line 18 is repeating line 16. Also, in the abstract and conclusion you mentioned that the technology has a moderate price, which contradicts on what is written on line 518.

Specific comments:

L16: delete the

L14-17: long sentence.

L18: is repetitive on what is again on line 22

L33: rewrite it, something like this: “an it was higher among Ho and CB than Jersey cows”

L43:Add reference

L40-43: long sentence

L43: Add reference

L45: how to instead of measures by which

L52: Severity instead of degree?

L53: add was after system

L54: Add and it after [36]

L56: delete description for and also

L58: what does reasonable reliability means?

L63: change < with less

L64: insufficient time for treatment for mild lameness?

L84: Such systems? Make reference to the locomotion scores?. If yes, I will add a ‘.’ After scores and write: These methods have been…

L86: delete not only and change used with uses

L87: delete but also

L88: data acquisition: is done by the camara per se?

L98: delete a 

L98: references

L105: the instead of those and delete of

L106: for instead of of

L117: You don’t need this sentence: the final search..

L123: techniques

L132: delete of

L144: delete The

L155: something is missing, there is a comparison however there is only one accuracy value reported.

L157: did you mean key features of lameness detection?

L166-170 this part is hard to understand

L170: were instead of are

L186: technique instead of technology – here and in the rest of the manuscript

L189: delete better

L189: improve instead of realize

L190: delete under the condition of and the

L197: delete characteristics

L207: team- is your lab, I will suggest that you write it in detail.

L218-219: How do you considerer healthy cows, with pericardial diseases, pleuropneumonia, ect?

L220-222: this is contradictory

L230: what does fusion judgement means?

L241: what do you mean by judge?

L248: experimentally calculated, how?

L249: statistically calculated, how?

L251: delete’,’ after complex

L251: individuals

L257: for instead of in

L258-271: this paragraph explains many details, not useful, for the purpose of this manuscript (see line 265 text in parenthesis). I will suggest emphasizing more in the results and the overall methodology.

L269: delete to

L270: add respectively after 91%,

L275: what is the idea behind this sentence?

L277: is complex, yes! But does it improve the prediction?

L287: This sentence don’t make sense- is really cow lameness or prediction models?

L288: add classified after correct

L288: Again, is not lameness detection- specificity, sensitivity and accuracy are metrics to evaluate models predictions is not that lameness detection refers to… rewrite it

L292: and instead of which

L314: delete by

L315: compared instead of comparison

L317: what do you mean by methods? Camera?

L317: add respectively after 95%,

L320: is repeated – already stated in L313

L325: Add reference

L341: Since is a patent and are results from your lab, you can add reference like Kang et al., unpublished…

L353: The speed is given by the image or is calculated?

L259: rewrite this sentence: …and can obtain images more easily

L363: is repeated

L378: Add reference

L388: what do you mean by statistical descriptions?

L391: Spell it out- No abbreviations starting a sentence

L416-419: to long

L418: delete methods after research

L423: do you mean decision-making of detection?

L428: delete and the milk industry

L429: delete and meaningful

L444-446: this sentence is too long and is hard to understand

L457: what is the difference between multifeature and multidata

L477: something is missing, detect basis what?

L496: DL is part of ML, so is not and

L502: how is research more convenient?

L511: In the literature? Or in this manuscript?

L516: add in after applied

L517: adoption instead of application

L518: the abstract and conclusion says otherwise

L528: Again, is not the computer vision per se detect the different health issues

L529: body weight?

L532: Add reference

L552: I think you mean that instead of there? If not, then what was the other problem?

L565: develop instead of realize?

L573: and instead to

L584: delete as a conclusion, we can say that

Table 1: the ‘/’ definition add it to the footnote of the table

I suggested to add a column with the statistical methods or algorithms, however what is written down for some of the studies are not statistical methods or algorithms, please review and if there isn’t any statistical methods or algorithms add NA or something similar.

Table 2: the ‘/’ definition add it to the footnote of the table

Same comment about the statistical methods or algorithms of Table 1.

What does it mean statistical analysis in the research method?

Change exceeds with was greater than

The font of the algorithm used is different

The font of the algorithm used is different in the Van Hertem et al.

The ROC is not an algorithm

Again, be consistent in how the results are reported in the tables, for example Bahr et al. is as “0.84” and the others are as percentage. Define KNN.

Fig.3

L238: (c and d) (e and f) – the ‘f’ is not bold

Reviewer 4 Report

The authors made appropriate changes to the paper for better readability and illustration. The paper makes a useful contribution in explaining and comparing current technologies of cattle health monitoring in general.
